# Self-assembly of highly ordered micro- and nanoparticle deposits

Hossein Zargartalebi[1,2], S. Hossein Hejazi [3] & Amir Sanati-Nezhad [1,2✉]

The evaporation of particle-laden sessile droplets is associated with capillary-driven outward flow and leaves nonuniform coffee-ring-like particle patterns due to far-from-equilibrium effects. Traditionally, the surface energies of the drop and solid phases are tuned, or external forces are applied to suppress the coffee-ring; however, achieving a uniform and repeatable particle deposition is extremely challenging. Here, we report a simple, scalable, and non-invasive technique that yields uniform and exceptionally ordered particle deposits on a microscale surface area by placing the droplet on a near neutral-wet shadow mold attached to a hydrophilic substrate. The simplicity of the method, no external forces, and no tuning materials' physiochemical properties make the present generic approach an excellent candidate for a wide range of sensitive applications. We demonstrate the utility of this method for fabricating ordered mono- and multilayer patternable coatings, producing nanofilters with controlled pore size, and creating reproducible functionalized nanosensors.

[1] Department of Mechanical and Manufacturing Engineering, University of Calgary, Calgary, AB T2N 1N4, Canada. [2] BioMEMS and Bioinspired Microfluidic Laboratory, Department of Biomedical Engineering, University of Calgary, Calgary, AB T2N 1N4, Canada. [3] Department of Chemical and Petroleum Engineering, University of Calgary, Calgary, AB T2N 1N4, Canada. ✉email: amir.sanatinezhad@ucalgary.ca

The evaporation of particle-laden sessile droplets deposited on a surface is often associated with contact line pinning, where particles move toward the edge and leave a ring-like pattern, which is referred to as a coffee-ring[1–4]. The irregular accumulation of particles at the contact line is primarily attributed to the outward capillary flow from the drop center, and the dried zone in the drop periphery is replenished as a result[3]. Despite the need for uniform particle coatings, the omnipresent nature of coffee-ring formation[4–7] creates challenges for many applications, including genotyping[8], diagnostics[9,10], and printing[11,12]. In nanobiosensing, for instance, it is very important to form an extremely uniform and controllable particle coating morphology to meet the high sensitivity required for sensors at various scales[13]. The physics involved in particle deposition depends on the shape[4,14–16] and type[17] of particles, the characteristics of the solution[3,7,18,19], the temperature[20,21], and wettability[22] of the surface substrate. In current practices, the motility of particles toward the droplet periphery is eliminated by imposing a counter flow inside the sessile droplet using electroosmosis[23,24], electrowetting[25], Marangoni flows[3,26,27], or dual-droplet inkjet printing[28,29].

Electrowetting of partially hydrophobic surfaces reduces the pinning effect and drags the contact line outward[30]. An electric field, which is applied between a circular electrode around the particle-laden droplet and a central point electrode, generates a radial inward electroosmotic flow, countering edgeward capillary flow, and consequently preventing ring formation[23]. In electrowetting, however, carefully manipulating the droplet at the center of the electrode and inducing a slight heating effect is necessary.

Augmenting the hydrophobicity of the surface is usually associated with a decline in the contact angle hysteresis, which leads to a decrease in contact line pinning[31–33]. The surface wettability can be controlled by chemical or physical treatments[32,34–36]. Hydrophobization of a porous surface is shown to be an effective physical approach to produce superhydrophobic surfaces, enabling the coffee-ring effect to be suppressed[37]. However, establishing a superhydrophobic surface is not only a costly process but also, in the cases of droplets containing surface-active materials, e.g., biological samples, deals with the strong pinning of the contact line. The change in the surface hydrophobicity promotes the adsorption of surface-active bioparticles. Given that the detachment of the nonspecific adsorption of these bioparticles is not a straightforward process. In summary, suppressing the coffee-ring effect through hydrophobization is still a challenge[38,39].

The cooling effect of evaporation induces a nonuniform temperature at the drop free surface, and the temperature is the lowest at the droplet's top center and the highest at the edge of the droplet. Consequently, there is a surface tension gradient on the free surface of the droplet; hence, the ring-like deposition of particles can be impeded or reversed by Marangoni stresses[3,40,41]. The Marangoni flow, stemming from the temperature gradient, is small in water droplets[3] but noticeable in volatile[42,43] and binary solutions[44]. Several approaches, such as local heating of the droplet surface using a laser beam[45], raising the substrate temperature[46], employing different solvents[11,47], adding surfactant or polymers to the droplet[7,12,18,27,48,49], drying a drop in an ethanol vapor atmosphere[42], and modulating the pH of the solution[50] have been developed to increase the Marangoni stresses. However, the process for intensifying the Marangoni flow may not be feasible in applications that necessitate either an intact particle-laden solution and/or highly ordered particle layers. For instance, the addition of nonconductive polymers and surfactants to suppress ring formation not only makes the sensors less conductive but also reduces their reproducibility.

Other approaches in controlling ring formation rely on flattening the droplet interface. This is achieved by placing the particle-laden drop on a second supporting drop, and this process is referred to as dual-droplet inkjet printing[28,29]. Using a pinning-depinning, employing confined geometries, such as curve-on-flat geometries[51], crossed cylinders of freshly cleaved single crystals of mica[52], and a mask[53], have shown effectiveness in suppressing capillary flow. The surface roughness also affects particles deposition, decreasing contact line pinning. Lateral drying can be reduced using a polydimethylsiloxane (PDMS) wall on the substrate[54]. Consequently, the particle accumulation is limited near the wall, and uniform deposition of particles on the internal area is achieved[54].

The most effective strategies for preventing ring formation involve heating or chemical dissolutions, which can be intrusive to the composition of the deposited layer. Moreover, the induced electroosmotic, capillary, and Marangoni flows cause a degree of nonuniform particle deposition and inversely affect the reproducibility and repeatability of the coated layers[18,48], which is crucial for extremely sensitive applications, such as sensors and electronics. Thus, there is a pressing need to develop a nonintrusive approach that can restrict the radial movement of particles.

The uniform deposition of salt (salt flat) on a bed of a dry saline lake, possibly due to the meniscus-free water-air interface offers an inspiring solution to address the challenge of ring formation. Translating the meniscus-free interface into microscale features enables eminent control of particle deposition.

Here, we demonstrate a state-of-the-art nonintrusive and repeatable method for the self-assembly of highly ordered particle layers with coating thicknesses greater than 100 nm and pattern lengths greater than 100 µm. The proposed technique is examined to fabricate highly repeatable and reproducible functionalized electrodes usable for making ultrasensitive nanosensors. Moreover, we demonstrate the self-stratification of a binary particle system that can spontaneously form filters with continuously increasing nanosized pores, ranging from 130 to 200 nm. A theoretical model reveals the key contributions of the substrate contact angle and the ratio of evaporation to particle diffusion time scales, which is defined as the Peclet number, in predicting the intricate interactions between particle deposition and evaporation. A general phase diagram is drawn to distinguish the regions of ring-free deposition in the plane of the contact angle-Peclet number. The specific features of the present coating technique, such as being noninvasive, independent of particle shape, fluid type, and surface morphology, and being easy-to-use, repeatable, and scalable, provide a universal methodology for coating and particle deposition to a wide range of applications in biology, advanced materials, and electronics.

## Results

### Nature-inspired meniscus-free and coffee-ring-free protocol.
We first present the conceptual model and steps for a meniscus-free evaporation arrangement to generate a uniform coating of particles. A salt lake acts as a stable colloidal gigantic film with a flat interface (Fig. 1a). When the lake evaporates, a layer of salt is coated on the lake bed, as there is no curvature at the air-water interface during evaporation (Fig. 1b). As such, the evaporation rate remains constant at the interface. When the evaporation rate is variable, radial flow leads to particle accumulation at the periphery or center of the drop (Fig. 1c). In the absence of Marangoni and capillary forces, the particles are gradually arrested at the descending liquid-air interface during evaporation and form a uniform deposition layer (Fig. 1d). The ring-free deposition of the salt in a salt lake is due to the large surface area covered by the

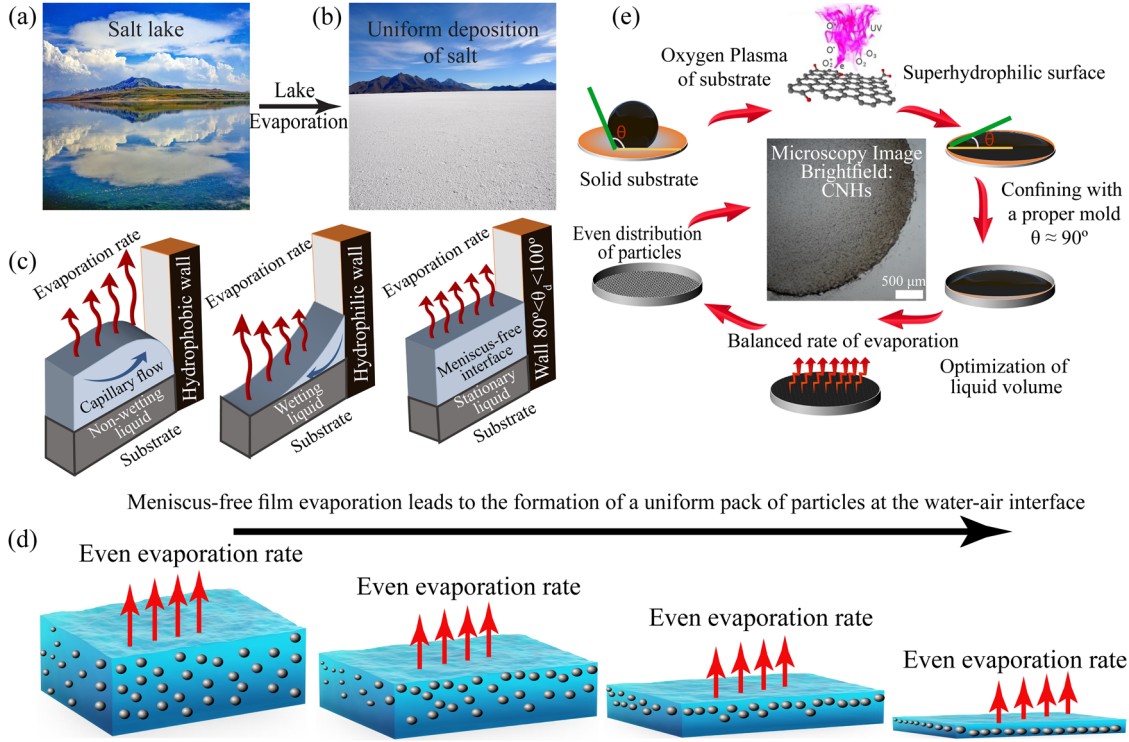

**Fig. 1 Nature-inspired uniform salt film formation in a salt lake. a** A salt lake represents a large uniform film with negligible curvature at the interface. **b** Salt deposited on the lake-bed after evaporation, showing a uniform coating. **c** The effect of the interface meniscus on the evaporation rate. For designs with a hydrophobic or hydrophilic interface meniscus, the closer the liquid-air interface is to the substrate, the higher the rate of evaporation and capillary flow. The meniscus-free interface of a mold with neutral wettability in our meniscus-free and coffee-ring-free protocol enables a coffee-ring-free particle deposition. **d** Meniscus-free film evaporation causes a gradual particle deposition at the liquid/air interface without any capillary flow, leading to the formation of a uniform layer. **e** The proposed protocol suppresses the coffee-ring effect by employing an oxygen plasma for substrate treatment and a shadow mold with neutral wettability.

water film. However, at the microscale, the radial movement of particles toward the drop edge due to the variable evaporation rate is significant, and controlling the interface curvature requires specific measures. Inspired by the nature of the salt lakes and scaling down to industrial use, a mold with a near-neutral wettability is mounted and attached to either a permanent or temporary hydrophilic substrate. A particle-laden drop is then placed on the hydrophilic surface surrounded by the shadow mold and is left to be dried. The meniscus of air-drop interface is eliminated, which consequently results in an even evaporation rate at the interface (Fig. 1e). Thus, a highly uniform deposition of particles is achieved (Fig. 1e).

To achieve uniform deposition, there must be both a meniscus-free interface and a hydrophilic substrate. The flat liquid-air interface eliminates the capillary flow and radial movement of particles. The hydrophilicity of the solid surface inhibits the thin film rupture; thus, it facilitates the formation of a liquid film from a very small drop volume, allowing for a descending interface to capture particles evenly during evaporation.

Here, we analyze the deposition of two different types of particles, (1) carbon nanohorns (CNH) with a diameter of 100 nm and zeta potential of −37.67 mV, which have broad applications in biosensors and capacitors[55], and (2) fluorescent and nonfluorescent spherical polystyrene-based particles of various sizes (with diameters of 200 nm, 1000 nm, and 10 µm) with established utility for studying the physics of coffee-ring phenomenon, stratification of binary particles, and formation of monolayers[1].

**The morphology of deposited particles**. The deposition of particles, laden in a drop, on a substrate suffers from the lack of a highly uniform pattern as the phenomenon is a complicated, far-from-equilibrium, and almost noncontrollable procedure[4,28,29,33,42,56,57] (Supplementary Fig. S1). Depending on the substrate wettability, particle migration changes from outward (coffee-ring) to inward (coffee-eye)[58]; therefore, film deposition is neither uniform nor replicable. To overcome the nonuniform coating of particles, a deposition spot is created by mounting a pressure-sensitive adhesive (PSA) mold with almost neutral wettability on a superhydrophilic glass substrate with a contact angle of 7° (Fig. S2 in the Supplementary Information). The hydrophilicity of the solid surface facilitates film formation from a very small drop volume, allowing the descending interface to capture particles evenly during evaporation. Evaporation from the meniscus-free surface forms a viscous-quasi-solid layer[59], preventing increased radial velocity of the particles at the final moments of evaporation (rush-hour)[60].

When there is no shadow mold, drop-casting on a hydrophilic glass slide leads to particle migration toward the pinned contact line, leaving the rings as expected (Fig. 2a and Supplementary Movie S1). However, the radial movement of CNH nanoparticles is suppressed when they are deposited on the surface, with the shadow mold forming a highly uniform coating (Fig. 2b and Supplementary Movie S2). It is noted that the wall's surface roughness may contribute to the interaction between the liquid film and the mold wall[61]. Therefore, the roughness of the PSA wall was measured using atomic force microscopy (AFM) (Fig. S3). The average roughness of the PSA wall was measured

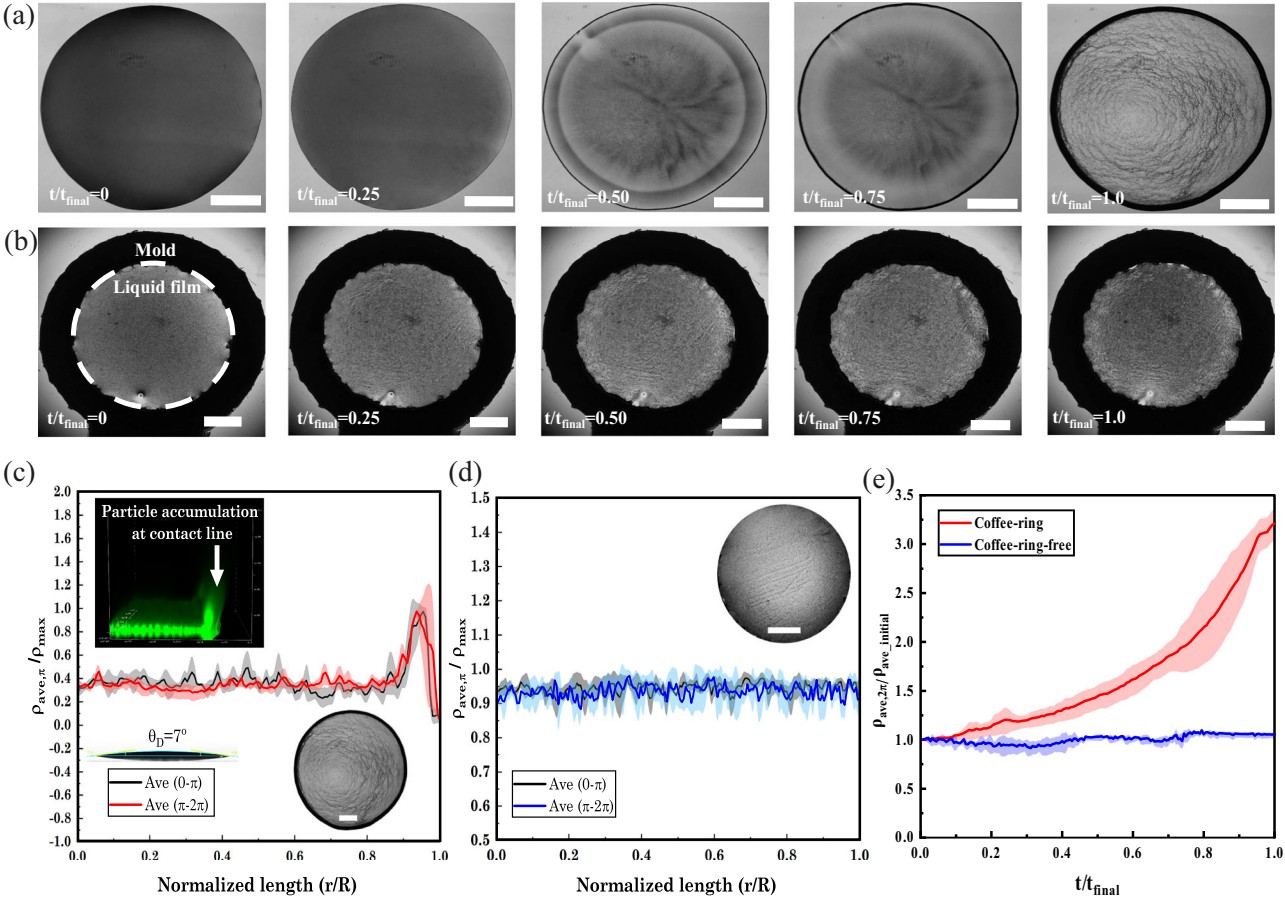

**Fig. 2 Coffee-ring versus coffee-ring-free nanoparticle deposition. a, b** Comparing the coffee-ring and coffee-ring-free particle depositions: experimental snapshots of the transportation of particles over time during evaporation. Scale bars in **a** and **b** are 500 and 1000 μm, respectively; The dark ring in **b** is the shadow mold (PSA layer). **c** The average density of deposited carbon nanohorn (CNH) nanoparticles with a size of 100 nm and zeta potential of −37.67 mV, dispersed in deionized (DI) water at a concentration of 0.5 mg/ml, on substrates with two different contact angles of 7° and 32°; scale bar: 500 μm. **d** Comparison of the average density of nanoparticles in two halves of the circular deposited layer using the proposed protocol, which shows the uniform coating of the particles; scale bar: 1000 μm. **e** Average distribution of the particles at the contact line over time is used to compare particle accumulation in the coffee-ring phenomenon and uniform particle deposition in the coffee-ring-free model. **c–e** Standard deviations are shown as shaded lines, with $n = 3$ for each case. Source data are provided as a Source Data file.

to be around 56 nm, which seems to have a negligible effect on the interaction of the liquid film and PSA wall, especially where the contact angle is ~90°. Studies have shown that surface roughness may become relevant at contact angles deviating from 90°, wherein the increased roughness of a surface intensifies its hydrophilic or hydrophobic property[62,63].

We quantify the spatiotemporal evaporation of drops for various morphologies of the deposited layers, as recorded in Fig. 2a, b, by calculating the normalized areal density of deposition as $\rho_{ave}(r)/\rho_{max} = 1/\rho_{max} \int \rho(r, \theta) d\theta$ versus normalized length ($r/R$) at $t/t_{final} = 1$, dividing the circular deposition area in two and evaluating the areal density for $[0, \pi]$ and $[\pi, 2\pi]$ (Fig. 2c, d). Without the shadow mold, a highly concentrated amorphous deposition is formed at the outer edge of the drop where $r/R > 0.8$ while the deposition remains almost uniform for $r/R < 0.8$, and this is mainly due to the small volume of drop and substrate contact angle (Fig. 2c and Supplementary Movie S3). We also show the densities measured for the nanoparticle deposition on a substrate with a contact angle of 32° (Supplementary Fig. S4), in which a strong radial flow results in a severe nonuniform deposition of particles. In contrast, forming a meniscus-free interface by shadow molding demonstrates that the average normalized densities for the two halves of the deposited layer are

similar (Fig. 2d). A uniform coating is achieved as long as the volume of the liquid drop is sufficient to fill the intended area surrounded by the mold. In all the above cases, the concentration of the suspended particles is low enough to prevent agglomeration at the liquid-air interface.

Moreover, we quantify the time evolution of particle deposition by measuring the average areal density of particles in a 20% area near the contact line as $\rho_{ave,2\pi}(t) = \int_0^{2\pi} \int_{R-0.2R}^{R} \rho(r, \theta, t) dr d\theta$. Note that $R$ represents the radius of the three-phase contact line in Fig. 2a (the case without the shadow mold) and decreases in time as evaporation continues. The same values of $R'(t)$ are used in the calculation of densities in both cases of Fig. 2a, b. $\rho_{ave,2\pi}(t)$ normalized by the initial areal density ($\rho_{ave\_init}$) are shown in Fig. 2e in black and blue for cases without and with shadow mold, respectively. While there was an amorphous accumulation of nanoparticles over time for a nonflat drop interface (the case without the shadow mold), the deposition became uniform for the meniscus-free interface (Fig. 2e). Quantitative monitoring of particle deposition over time shows that the normalized $\rho_{ave,2\pi}$, calculated for the coffee-ring effect, significantly deviates from one, i.e., increases up to three times, while it remains close to one for the case with mold. This confirms the reliability of the

meniscus-free protocol with respect to eliminating the ring pattern (Fig. 2e and Supplementary Fig. S5).

Microparticle image velocimetry (μPIV) was used to visualize the motility and deposition of particles during the liquid film evaporation under the no-coffee-ring condition. The results confirm the absence of radial motility of particles during their deposition (Supplementary Fig. S6). Tracing of particles during their deposition shows that the net radial movement of particles is almost zero (Fig. S6a), although they have small random fluctuations during evaporation (Fig. S6b).

We also examined whether the wall shape (circular, rectangular, or A-like patterns) affects particle movement during evaporation. The results of μPIV and brightfield microscopy analysis show that the method has generality considering that the local evaporation gradients, which stem from the noncircular curvature of the wall, have a negligible effect on the evaporation rate (Supplementary Movies S4 and S5 and Supplementary Fig. S7).

**Deposition of monolayered microparticles**. The capability of the meniscus-free technique in forming a monolayer of particles was examined using polystyrene microparticles (10 μm diameter and −36.5 mV zeta potential). The particle-laden drop concentration was set to 2.7% w/v, which is equivalent to ~$4.914 \times 10^7$ particles/ml. This is calculated according to the surface area of the deposited layer, as a higher concentration of microparticles can potentially form multilayer depositions. The microparticles are deposited on a glass slide with mounted shadow mold, and the uniformity and thickness of the deposited layer are evaluated through brightfield microscopy and AFM. Although depositing these particles using the conventional drop-casting method on a glass substrate culminates in an irregular deposition layer (Supplementary Fig. S8), Fig. 3a illustrates the arrangement of microparticles in the form of a monolayer inside the shadow mold. The magnified images of the contact line show no sign of the coffee-ring effect.

The AFM topography also verifies the uniform patterning of these microparticles (Fig. 3b). To assure the monolayer formation, the height of the deposited layer was measured using JPK data processing software 6.1.111, in which the local uniformity of the measured height confirms the monolayer deposition of microparticles (Fig. 3c). The average roughness ($R_a$) and root mean square roughness ($R_q$) were calculated at five different areas ($100\,\mu m \times 100\,\mu m$) of the deposited layer to evaluate the uniformity of the particle distribution in the entire deposited surface. The measured roughness values at different spots, in Fig. 3d, are very close with $\Delta R_{a,max} = 2.2\%$ and $\Delta R_{q,max} = 5.2\%$, indicating uniform microparticle deposition independent of the selected area.

**Patterning and stratification of particles to form filters with nanosized pores**. A key feature of the present protocol is the flexibility in the design of the shadow mold to noncircular geometries. Hence, we can create coatings with desired patterns as necessary in solution-processable electronics, biotechnology, and combinatorial chemistry[64,65]. For illustration purposes, a PSA mold with a tree-leaf design is mounted on the hydrophilic substrate with a contact angle of 7°. The total volume fraction of particles is $\varphi_0 = 0.01$, and the fluid volume is 6 μl which can form an initial film thickness of ~$H = 500\,\mu m$ over the desired area. We captured the 3D scanning of the deposited particles using confocal microscopy. The water drop spreads and covers the pattern area, as shown in Fig. 4a. The enlarged images of the leaf pattern in Fig. 4a depict the existence of the liquid film in all narrow and corner points of the leaf.

We placed a drop of water containing two types of fluorescent polystyrene particles 200 nm diameter (referred to here as fine particles and shown in green) and 1000 nm diameter (referred here as microparticles and shown in red) and the area bounded by the shadow mold (Fig. 4b). The bimodal dispersed particles are spontaneously stratified in the designed pattern during evaporation with the deposition of mainly green particles (200 nm diameter) at the solid surface, which gradually transitions to red particles (1000 nm diameter) on the top (Supplementary Movie S6).

It is noted that the surface coating at a low concentration of polymer particles may not be highly uniform (Fig. 4b), owing to the possible local thermal/solutal gradients of the interfacial tension along the water-vapor interface of the evaporating film[66–68]. Increasing the concentration of particles helps to cover a larger surface of the liquid/air interface and improves the uniformity of the deposited layer formed (Fig. 2b, d). In regard to particle stratification, increasing the concentration of particles reduces the free movement of particles during their deposition and increases the transition zone, owing to the increased particle-particle interaction (Supplementary Fig. S9a). However, the stratified particles can make highly uniform multilayer films with no defects in the deposits (Supplementary Fig. S9b).

We evaluated the fraction of green and red particles along the height of the deposited layer from the 3D confocal images (Fig. 4c), in which the fraction of green (fine) particles at the bottom was three times greater than that of red (large) microparticles. Moving upward along the normalized height ($h^*$), this ratio reaches unity around $h^* = 0.2$. Afterward, the fraction of the large microparticles surpasses the small ones, where the ratio reaches half. There is a transition zone in between ($0.15 < h^* < 0.3$) wherein the volume fractions of small and large particles are comparable. Mechanistically, the suspended particles undergo Brownian motion and cause the colloidal suspensions to reach equilibrium. On the other hand, the descending liquid-air interface during evaporation enforces the particles to come into close-packing at the interface. The competition between these two phenomena induces the self-stratification of particles[69,70]. Stratification behavior might also be affected by cross interactions among the particles[71] and the evaporation ratio to the Stokes-Einstein diffusion time scales, defined as the Peclet number[69,70]. In other words, the Peclet values of the large and small particles determine the final composition of the deposited layer. For instance, if the Peclet values of both small and large particles are larger than one, the small particles segregate in a layer on top of the larger particles. However, if the Peclet values of the large and small particles are respectively higher and smaller than one, the small particles form as the bottom layer[69,70].

Three-dimensional monitoring of the self-organization of particles over time in a drying film can elucidate the distribution pattern of particles of different sizes along the height (Supplementary Fig. S10). The volume fraction of particles (suspended in the film) in the early steps of film evaporation is much less than that of the final stages. In the later stages of evaporation, the particles are highly concentrated, suppressing the occurrence of diffusional organization. The locations of particles and the air-water interface (shown qualitatively in Supplementary Fig. S10) are determined and shown in a heatmap (Fig. 4d). The heatmap depicts the stratification of particles over nine time intervals. The ratio of fine (200 nm) to large (1000 nm) particles in the liquid film varies during liquid evaporation (Fig. 4d). Initially, both fine- and microparticles are evenly distributed in the film when the normalized time equals 0.11. As the liquid-air interface descends, the 200 nm particles tend to move down faster than the large-1000 nm particles due to their higher diffusion rate when

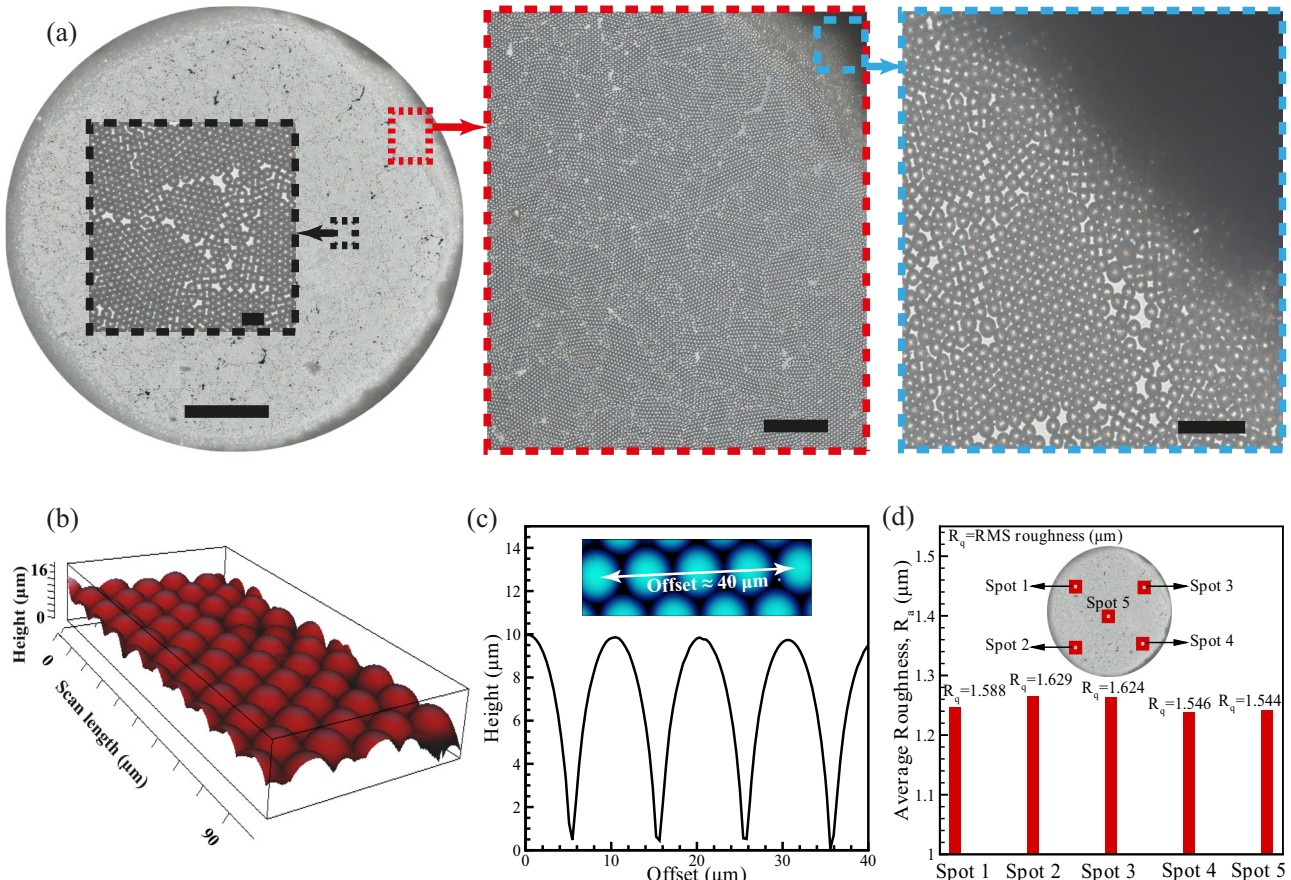

**Fig. 3 Monolayer deposition of the polystyrene microparticles using the proposed method. a** Brightfield microscopy image of the deposited microparticles, representing monolayer formation along the surface and at the contact line; scale bars: left (main: 500 μm, inset: 20 μm), middle: 200 μm, right: 50 μm. **b** Three-dimensional (3D) view of the deposited microparticles using atomic force microscopy (AFM) topography image, showing highly ordered microparticles. **c** The measured height of the deposited layer for five microparticles placed next to each other, analyzing the AFM image (offset is the length of the sketched line). **d** The average roughness ($R_a = 1.25 \pm 0.012$, 95% confidence interval) and root mean square (RMS) roughness ($R_q = 1.59 \pm 0.04$, 95% confidence interval) were measured at different spots of the deposited layer, confirming the uniform distribution of microparticles. Source data are provided as a Source Data file.

$0.11 < \tau < 0.44$. However, the discrepancy in downward particle velocity decreases when $\tau > 0.44$ as the particles in the liquid film become highly concentrated and have a limited space to move (Fig. 4d). The pore size (PS) of the deposited layers is calculated with the assumption that the particles form a uniform packing. We calculate the variation in average PS along the height of the deposited layer as $\gamma\left[\alpha_{sl} \times \mathrm{PS}_s + (1 - \alpha_{sl}) \times \mathrm{PS}_l\right] + (1 - \gamma)\mathrm{PS}_f$, where $\alpha_{sl}$ is the volumetric ratio of 200–1000 nm particles along the height as evaluated in Fig. 4c and $\gamma$ is the particle fraction of each scanned layer throughout the height of the deposited film. Assuming a uniform arrangement of particles, $\mathrm{PS}_s$ and $\mathrm{PS}_l$ are ~46 and 231 nm, respectively. $\mathrm{PS}_f$ is the average PS of the observed fractures, calculated for each scanned layer throughout the film height. The value $\gamma$ is used to calculate a weighted average of close-packing particle PS ($\alpha_{sl} \times \mathrm{PS}_s + (1 - \alpha_{sl}) \times \mathrm{PS}_l$) and fracture radius ($\mathrm{PS}_f$)[72]. The PS has the smallest value of 120 nm for the close-packing situation and 212 nm for the less dense packing case at the solid surface. The PS gradually increases to 200 and 386 nm, respectively, at the top of the coated layer (Fig. 4e). The shaded area highlights the PS difference between the close-packing system and the less dense one. With the increase in particle concentration, $\gamma$ value tends to reach unity and the average PS approaches the close-packing PS. The PS distribution reveals an interesting capability of the present protocol to fabricate microfilters whose PS are on the nanometer

scale and gradually increase along the cross-section of the filter (Fig. 4e). Filters with nanosized pores can be adopted for various applications where filtering with different sizes, patterns, and porosities is desirable. Although it is shown with one set of particles with two distinct sizes, fabricating customized microfilters with desired PS variations is feasible through tuning the size ratio and concentration of particles, film thickness, and evaporation rate.

**The theoretical model and phase diagram.** We develop a simple theoretical model to explain the physics of the evaporation process from a liquid film bound by a shadow mold and the mechanisms underlying the various deposition patterns. The model shows that various deposition regimes can be exclusively determined by the Peclet number, the confined contact angle of the film with the mold, and the pinning time.

Given the negligible gravitational force on a confined liquid surrounded by a microscale mold (with a Bond number less than one), the liquid-air interface approximately adopts a spherical cap shape when the wettability of the mold is not neutral. Considering the height of the formed film $H = H_1 + H_2$, the hydrodynamic radius $R$, and the contact angle $\theta$ (Fig. 5a) and assuming that pinning starts at the equilibrium magnitude of the contact angle, the normalized pinning time associated with each contact angle range can be derived by calculating the evaporation rate

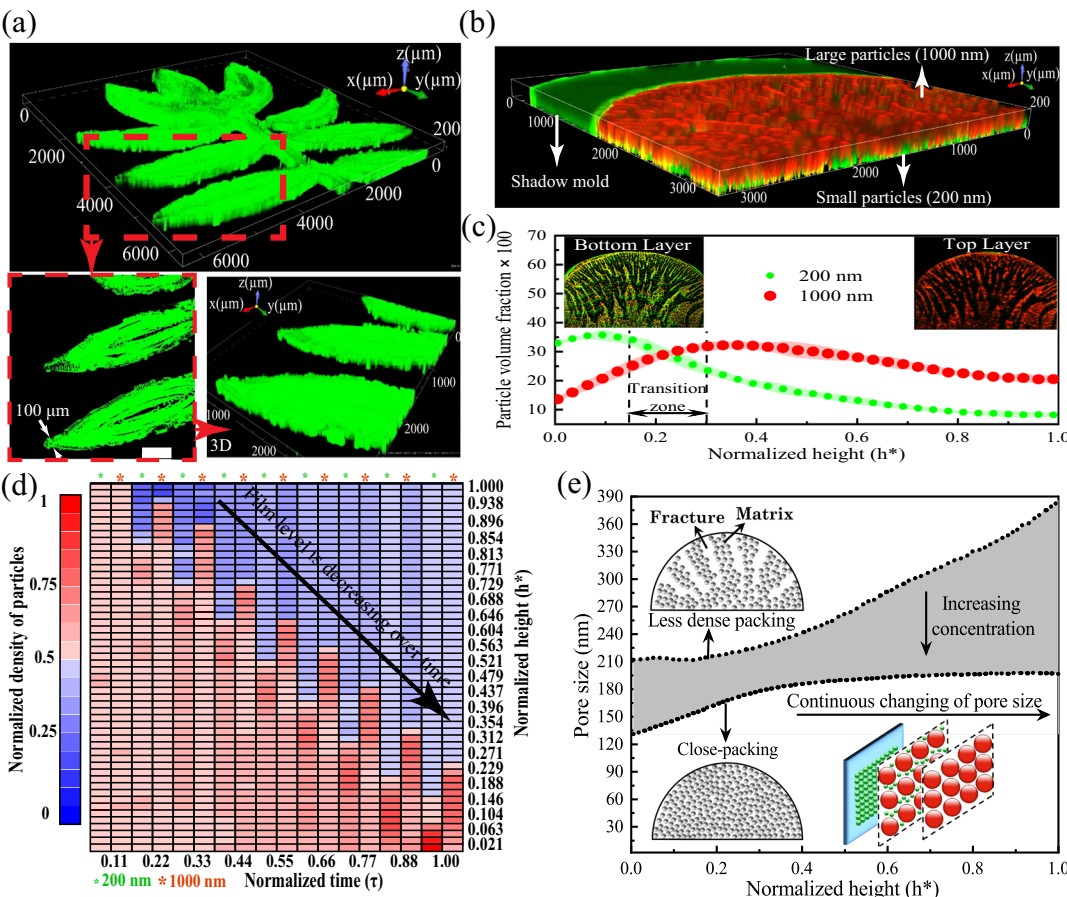

**Fig. 4 Patterning and self-stratification of fine- and microparticles using the proposed protocol. a** Uniform patterning of fluorescent polystyrene fine particles within a microscale patterned mold on a glass slide; scale bar: 500 μm. **b** Patterning a multilayer nanocomposite consisting of two different particle sizes by self-organization of these particles as a result of different diffusion rates, demonstrating the capability of the protocol for patterning multifunctional layers. **c** Volume fraction of the fine- and microparticles versus normalized height shows the high intensity of the small fine particles (200 nm) at the bottom layers and an opposite trend for the large microparticles (1000 nm). Standard deviations are shown as shaded lines, with $n = 3$. **d** Heatmap of the deposition of the mixed particles over time, revealing the autostratification of the particles. **e** The average nanoscale pore size of the deposited fine- and microparticles, in close-packing and less dense packing states, shows the pore size gradient along with the layers. The shaded area represents the pore size range in higher concentrations of the particles with respect to the less dense packing case. Source data are provided as a Source Data file.

(see Supplementary Information for the details of derivation):

$$\left|\frac{t_{Pin}}{\triangle t}\right| = \begin{cases} \frac{H}{R}\frac{\delta\theta}{P_e(1+\sin\theta)^2} & \theta > \frac{\pi}{2} \\ \frac{H}{R}\frac{\delta\theta\sin\theta}{P_e(1+\sin\theta)^2} & \theta < \frac{\pi}{2} \end{cases} \quad (1)$$

where $\triangle t = t_{Adv} - t_{Diff}$, in which $t_{Adv}$ is the time the particles migrate to the peripheral section and $t_{Diff}$ denotes the time the particles ward off the advective flow due to the build-up of particles at the contact line and the development of the particle concentration gradient. The contact angle variation during pinning ($\delta\theta$) depends on the excess Gibbs free energy and potential energy barrier (details in Supplementary Information). $Pe$ is the Peclet number and shows the magnitude of evaporation to the Stokes–Einstein diffusion timescales as $Pe = \dot{E}H/D_{diff} = (6\pi\eta r_p\dot{E}H)/k_BT$, where $k_B$ is Boltzmann's constant, $T$ is the solvent temperature, $\eta$ is the viscosity of the solvent, $r_p$ is the hydrodynamic radius of the particles, and $\dot{E}$ is the evaporation rate of the solvent. $H$ in Eq. (1) is the initial film height which is defined as $H_1 + H_2$. As increasing the hydrophobicity is associated with decreasing the contact angle hysteresis, and consequently decreasing the contact line pinning[31–33], the $\delta\theta$ value can be generally estimated as

$\delta\theta \approx a/\theta^b$, where a and b are the positive constant values depending on the physical and chemical conditions of the substrate. For a case where $a = b = 1$, a general map presenting the normalized pinning time, $\left|t_{Pin}/\triangle t\right|$, as a function of the Peclet number and contact angle for different values of $H/R$ (Fig. 5b) is plotted. The $Pe$ and $\theta$ parameters inherently contain the overal interactions of evaporation rate, solvent composition, film thickness, particle size, and capillary pressure. For a superhydrophilic substrate with a much higher wetting affinity than that of the mold, it is assumed that the pinning tendency of the contact line to the substrate is much greater than that of the mold.

The deposition is uniform when $\left|t_{Pin}/\triangle t\right| \ll 1$ and is otherwise nonuniform. The phase diagram of pinning time in Fig. 5b shows three distinct deposition signatures of particles based on $Pe$, $\theta$, and $H/R$, which are ring-like (coffee-ing), bell-like (coffee-eye), and ring-free (uniform). For $\theta > \pi/2$, the deposition forms coffee-ring patterns. This region shrinks as $Pe$ increases or the contact angle shifts toward $\pi$. Similarly, for contact angles at or close to 0, coffee-eye patterns are formed, which become more uniform as $Pe$ increases or $\theta$ decreases toward $\pi/2$. The regions of coffee-ring/eye-ring patterns become wider as $H/R$ increases.

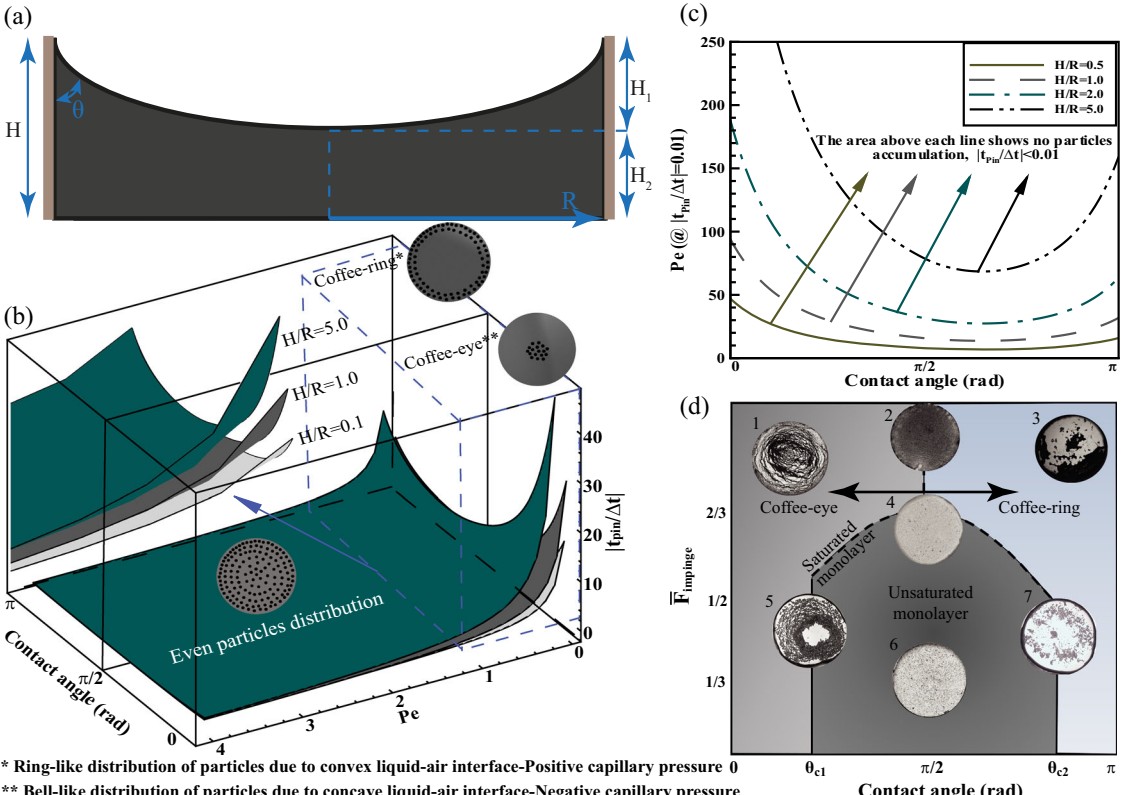

\* Ring-like distribution of particles due to convex liquid-air interface-Positive capillary pressure

\*\* Bell-like distribution of particles due to concave liquid-air interface-Negative capillary pressure

**Fig. 5 Phase diagram for confined particle-laden film evaporation. a** Schematic of the theoretical model, where $R$ is the hydrodynamic radius of the mold, $\theta$ is the contact angle ranging from 0 to $\pi$, and $H = H_1 + H_2$ is the height of the film. **b** 3D phase diagram of different regimes of particle deposition for various $H/R$ ratios, in which the distribution of particles on a superhydrophilic solid surface is dependent on the Peclet number of particles ($Pe$) and $\theta$. The $\theta$ value represents the deposition morphology, where it creates a ring-like structure when $\theta > \pi/2$ and a bell-like structure when $\theta < \pi/2$. Low values of $Pe$ intensify the accumulation of particles because of the higher diffusion rate of particles. An even particle distribution occurs when $Pe$ is high, and $\theta$ tends to $\pi/2$. **c** The $Pe$ number as a function of $\theta$ for $|t_{Pin}/\triangle t| < 0.01$, assigning the no-coffee-ring and no-coffee-eye area (safe zone) at different $H/R$ ratios. **d** 2D phase diagram along with experimental data showing the monolayer, coffee-ring, and coffee-eye areas as a function of $\bar{F}_{\text{impinge}}$ and $\theta$. The $\bar{F}_{\text{impinge}}$ and $\theta$ values for the experimental data are (1) $\bar{F}_{\text{impinge}} \approx 0.72$, $\theta \approx 54°$; (2) $\bar{F}_{\text{impinge}} \approx 0.9$, $\theta \approx 90°$; (3) $\bar{F}_{\text{impinge}} \approx 0.86$, $\theta \approx 106°$; (4) $\bar{F}_{\text{impinge}} \approx 0.6$, $\theta \approx 90°$; (5) $\bar{F}_{\text{impinge}} \approx 0.42$, $\theta \approx 54°$; (6) $\bar{F}_{\text{impinge}} \approx 0.35$, $\theta \approx 90°$; and (7) $\bar{F}_{\text{impinge}} \approx 0.5$, $\theta \approx 106°$. Source data are provided as a Source Data file.

The coffee-eye distribution of particles is attributed to the negative capillary pressure ($\theta < \pi/2$), which induces a concave upward liquid-air interface, leading to particle migration toward the center of the liquid. When the capillary pressure is positive ($\theta > \pi/2$), less pinning occurs on the mold wall, but the convex interface in the confined area leads the peripheral contact line to reach the substrate. Outward particle migration is enforced, resulting in the formation of a ring-like morphology. The zero-capillary pressure ($\theta = \pi/2$) is interpreted as a meniscus-free interface and uniform deposition of particles. Nonetheless, increasing the Peclet number decreases the tendency of particles to diffuse away from the descending interface and increases the chance of forming pattern-free (safe zone) in the nonzero capillary pressure. As expected, the safe zone extends when the Peclet number increases. A high $|t_{Pin}/\triangle t|$ occurs for large negative or positive values of the capillary pressure, where the interface of the evaporating film touches the solid substrate from the center or peripheral zones, respectively, creating a pinned contact line. This causes particle accumulation at the contact zone in the early- or mid-steps of film evaporation. Similarly, when the Peclet number is small, the particles easily diffuse away from the interface, which therefore induces a high $|t_{Pin}/\triangle t|$ (Fig. 5b). To avoid any local accumulation of the particles, the pinning time ($t_{Pin}$) needs to be much smaller than $\triangle t$. Therefore, the safe zone area, where $|t_{Pin}/\triangle t| < 0.01$, can be determined through $Pe$ and $\theta$,

and it changes as a function of $H/R$ ratios (Fig. 5c). Decreasing $H/R$ nullifies the edge effect and increases the safe zone area (Fig. 5c).

The regions of coffee-ring, coffee-eye, and monolayer depositions can also be distinguished by evaluating the normalized impingement flux of particles on the interface according to Eq. (2):

$$\bar{F}_{\text{impinge}} = \frac{1}{3}\frac{N_A r_p^2}{R^2}(1 + \sin\theta) \qquad (2)$$

where $N_A$ is the number of particles at the interface. The three deposition signatures on the plane of $\left(\bar{F}_{\text{impinge}}, \theta\right)$ reveal that monolayer deposition occurs in a region when $\theta_{c1} < \theta < \theta_{c2}$ and bounded by a dashed line when $N_A r_p^2/R^2 \leq 1$ (Fig. 5d). $\theta_{c1}$ and $\theta_{c2}$ are dependent on the $H/R$ ratios and $Pe$ values (Fig. 5b and Supplementary Fig. S11) where $|t_{Pin}/\triangle t| < 0.01$. $N_A r_p^2/R^2 = 1$ represents a substrate fully saturated with particles, named the saturated monolayer. The values of $\theta_{c1}$ and $\theta_{c2}$ lead to $\pi/2$ when $H/R$ decreases, meaning that the monolayer formation for large areas is much easier than that for small areas (Supplementary Fig. S11).

For the case of CNH nanoparticle deposition in the previous section, $P_e \approx 33$, $\theta \approx 95°$ for the PSA mold, and $H/R = 2.0$, we reach $|t_{Pin}/\triangle t| = 9.1 \times 10^{-3} \ll 1$. Thus, the theoretical model accurately predicts the highly uniform deposition of the CNH

nanoparticles, as shown in Fig. 2b. Similarly, for the 200 and 1000 nm polystyrene particles, the $|t_{Pin}/\triangle t|$ values are equal to $4.6\times 10^{-3}$ and $8.9\times 10^{-4}$, respectively, indicating the ring-free deposition. It is worth-mentioning that, in the case of a mold with a contact angle <85° and considering the similar parameters mentioned above, $|t_{Pin}/\triangle t|$ for the CNH nanoparticles exceeds the threshold and becomes higher than 0.01, representing the initiation of particle accumulation at the center of the substrate.

The patterns of particle deposition and the impinging flux were examined at different wettability levels of the mold wall to assess the extent of correlation of the theoretical model and experimental data. PDMS molds with tunable wettability (treated under oxygen plasma)[73] but identical roughness were used for this set of experiments. Plasma treatment renders hydrophilicity to an initially hydrophobic PDMS. The experimental time is <20 min ensuring that the change in the PDMS contact angle was negligible. Figure 5d shows the patterns of particle deposition for various contact angles and impinging fluxes. The patterns of particles indicate that the coffee-eye occurs when the wall contact angle is lower than the optimal value of 90°. For the plasma-treated (hydrophilic) PDMS wall (here with $\theta = 54°$), liquid break-up may occur, which leads to the creation of a possibly random hollow feature (uncoated) in the middle of the substrate (case 5 in Fig. 5d). In the case of a hydrophilic wall, although some particles were observed attaching to the PDMS wall, the concentration of these particles was negligible compared to the concentration of particles deposited on the substrate. For systems with $H \gg R$, however, this assumption may not be valid. For the intact PDMS molds with a hydrophobic wall (here with $\theta = 106°$), a high percentage of particles deposited on the edge of the substrate (cases 3 and 7 in Fig. 5d), as confirmed by the theoretical model.

## Discussion

We utilized a passive protocol to suppress the coffee-ring effect, which is applicable to uniform film formation at both the micro- and nanoscales. To halt the radial movement of particles and eliminate ring formations, the drop curvature needs to be flattened by utilizing a combination of a superhydrophilic substrate along with a neutral-wetting mold with low roughness. For such a system, we demonstrate that the particles are orderly located in the film and uniformly packed when the liquid-air interface is vertically receding during film evaporation. Hence, a highly organized monolayer or multilayer of the particles were deposited on the target location. We also investigated the mechanisms underlying the stratification of a colloidal drop with two different particle sizes. Based on the theoretical and experimental data, it is concluded that in the presence of a mold with a near-neutral-wetting angle, a larger particle size, a smaller ratio of the liquid film thickness to the target area for deposition, and a higher evaporation rate positively contribute to the formation of a highly uniform particle coating on either a permanent or temporary hydrophilic substrate, and beyond this, the particles accumulate in either ring-like or bell-like morphologies.

Eliminating the drop curvature during evaporation introduces nonintrusive coating and patterning approaches for applications that desire a highly uniform and controllable deposition of micro- and nanoparticles. Given the use of a mold in this work, the proposed protocol can be used for coating layers of particles in different patterns and has the potential for scalable production of uniform coatings. The deposition of binary polystyrene particles revealed the possibility of particle stratification for nanofilter fabrication purposes. However, reaching a highly uniform nanofilter structure, made of two or more than two particles sizes,

requires more investigation in terms of appropriate particle concentrations ratios.

It is also concluded that the theoretical model, supported by the data from experiments, could reasonably predict the coffee-ring/eye behavior presented in this work when the wall contact angle deviates from 90°. However, other parameters such as particle attachment to the wall and liquid film rupture may also contribute to changing the final pattern of particles, which needs to be evaluated by further experimentation.

Comparing the advantages and drawbacks of different coffee-ring suppression methods (summarized in Supplementary Table S1), the present technique is a promising approach for highly ordered, dry-mediated self-assembly of particles. It is easy-to-use, low-cost, scalable, and additive-free and has no dependency on particle shape or substrate topography. This method also enables the creation of uniform coating patterns of particles down to microscale feature sizes and in any arbitrary form.

## Methods

To address the needs of the proposed method to avoid or minimize the meniscus at the air/liquid interface, the mold needs to be made of a material with a contact angle of ~90° ± 10°. The substrate is not restricted to a particular material as far as it is highly hydrophilic ($\theta < 10°$, with or without treatment). The mold fabrication and substrate preparation protocols are detailed below.

**Mold fabrication**. Pressure-sensitive adhesive (PSA) material (biocompatible double-sided tape, Adhesives Research, Inc., US) is widely used in the scalable production of medical devices and has an almost neutral wettability ($\theta = 90° \pm 5°$). Therefore, PSA was selected as a qualified material for making shadow molds in this work (Supplementary Fig. S2). To pattern the PSA sheets (ARcare® 90106NB, 140 µm thick), the PSA was laser cut through a $CO_2$ laser cutter (Trotec SP500, Australia), set with power = 40%, speed = 12.7 cm/s, and PPI/Hz = 1000, which is capable of generating features down to 100 µm, according to the designed CAD file sketched in AutoCAD software (Autodesk, USA). The laser-cut PSA was adhered to the substrate (e.g., here glass slide, model: 1402, Globe Scientific Inc., U.S.; and gold screen-printed electrode, model: SPE250, Metrohm, Canada) using its adhesive layer to form the mold. The PSA mold with neutral wettability created a zero meniscus for the air/liquid interface, minimizing the capillary and Marangoni forces and preventing radial migration of particles. Notably, the PSA could act as a temporary mold and be easily peeled off from the substrate following the completion of particle deposition. Depending upon the drop volume and desired particle-laden film thickness, the mold thickness can be adjusted by stacking multiple PSA layers.

**Hydrophilic treatment of the substrate**. To prepare a hydrophilic substrate needed for the proposed method, an oxygen plasma treatment was applied (plasma machine: Electro-Technic products Inc, 50 W, 1 min) on the target untreated substrates (e.g., glass slides and electrodes) (Supplementary Fig. S2). As plasma treatment is known for cleaning substrates and increasing their wettability, many other types of materials can also be used as substrates in this method. Additionally, there is no need for surface treatment for materials with an intrinsically hydrophilic property ($\theta < 10°$).

**Microscopy imaging of particles during deposition on substrates**. The 3D images of the deposited films and the dynamics of fluorescent polystyrene particles (200 and 1000 nm sizes) during their coating were captured by confocal microscopy using a green laser with a wavelength of 488 nm and a red laser with a wavelength of 561 nm (Fig. 4a, b and Supplementary Figs. S1, S9 and S10). Long working distance lenses X2 and X4 Nikon microscopes were utilized to capture the images. Brightfield microscopy (model: DS-Ri2, Nikon Instruments) was used to capture the deposition dynamics of functionalized CNH nanoparticle (Sigma-Aldrich; Cat No., 804126) on glass slide substrates (Fig. 2c, d and Supplementary Movies S1–3 and S6). The topography of the deposited particles, the height of the formed monolayer films, and the roughness of the PSA mold were measured using AFM (Nanowizard 4 XP Nanoscience, Bruker, USA). AFM was used in tapping mode using a Multi75AI cantilever.

**Electrochemical sensing meets the proposed method**. While imaging techniques (AFM and confocal microscopy) were used to evaluate the dynamics and microscale uniformity of the deposited layers, we also took advantage of electrochemical sensing to investigate the microscale uniformity and repeatability of the deposited layers. On the other hand, the utility of the method for developing highly conductive and reproducible functionalized electrodes useful for making ultra-sensitive nanosensors was demonstrated.

The gold screen-printed electrodes (model: SPE250, Metrohm, Canada) were first electrochemically cleaned by applying a cyclic scanning to a potential range of 0 to 1.6 V in 0.5 M $H_2SO_4$ (Sigma-Aldrich; Cat No, 339741) at a scan rate of 100 mV s$^{-1}$ using a PGSTAT204 Potentiostat/Galvanostat (Metrohm-Autolab, the Netherlands) instrument. CNH nanoparticles were then drop-cast on cleaned gold screen-printed electrodes under the proposed protocol discussed above. The differential pulse voltammetry method (with the help of Nova 1.11 software), known for characterizing infinitesimal changes in deposited nanomaterials, was employed to quantitatively evaluate the uniformity and repeatability of the CNH coatings (Supplementary Fig. S5). To measure the DVP electrical current signal, the coated electrodes were rinsed with DI water to eliminate contaminants. They were then immersed in a PBS solution (0.02 M, pH = 7.4) consisting of the redox probe (2.5 mM potassium ferricyanide ($K_3Fe(CN)_6$, Cat No, 702587, Sigma-Aldrich; and 2.5 mM potassium ferrocyanide ($K_4Fe(CN)_6$, Cat No, P3289, Sigma-Aldrich) and subject to a potential range of $-0.4$ to 0.6 V with a scan rate of 100 mV s$^{-1}$ (Fig. S5).

**Zeta potential measurement and stability analysis of particle suspensions**. To ensure that the effect of gravity on the suspended nano-, fine-, and microparticles is negligible and the particles stably remain suspended during the sample preparation and evaporation period, we measured the zeta potential for both functionalized CNH (100 nm, see below) and polystyrene fine- and microparticles (200 nm, 1000 nm, and 10 μm). We developed and optimized a protocol to functionalize CNH nanoparticles to enhance their stability in DI water. Based on this optimized protocol, 5 ml hydrochloric acid (HCL) (Sigma-Aldrich), 10 ml sulfuric acid ($H_2SO_4$) (Sigma-Aldrich), and 5 mg CNH nanoparticles were mixed and heated for 72 h in a hot silicone oil bath (120 °C) over a Corning™ Pyroceram™ Hot Plate Stirrer. Upon cooling down after 1 h at room temperature, the functionalized CNH nanoparticles were then washed with DI water three times for neutralization and to achieve pH≈ 7.0, followed by drying in a vacuum oven (120 V, 1600 W, Thermo Scientific™ 36185) at 70 °C for 12 h. Dynamic light scattering (DLS), Nanoseries ZS, Malvern Instrument, was used to measure the zeta potentials of the functionalized CNH and polystyrene particles. According to the DLS results, the zeta potentials of the CNH and polystyrene particles were measured to be $-37.67$ and $-36.5$ mV, respectively, representing their high stability in DI water and demonstrating the negligible effect of gravity on particles. It is noted that the particles used in this study remain stable in their stock solutions; therefore, no precipitation of these particles occurred in weeks of suspension. Moreover, considering the evaporation rate of 0.1 mm/min[9] at room temperature (25 °C), using Stokes law, the gravitational time-scale for 1000 nm polystyrene particles in a 10 mm water film, neglecting the particle charge ($t_g = \frac{9h}{2gr_p^2}\frac{\mu}{(\rho_p - \rho_f)} \approx 9 \times 10^5$ s) is a few orders of magnitude higher than the evaporation time-scale ($t_E = \frac{h}{E} = 6 \times 10^3$ s). This time-scale analysis shows that the effect of gravity is negligible with respect to the evaporation rate[9].

## Data availability
Source data are provided with this paper.

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

## Acknowledgements

We would like to acknowledge Dr. Mohammad Zargartalebi at the University of Toronto for his valuable advice in improving the quality of the paper. We also appreciate Tom Till/Alamy Stock Photo and Alex Neumayer/Alamy Stock Photo for sharing Fig. 1a, b, respectively, through the Alamy website. This study was financially supported by the Natural Sciences and Engineering Research of Canada (NSERC), discovery grant No. 10010499; Canadian Institutes of Health Research, Canada, grant No. 10028463; and Canada Research Chair, grant No: 10027532 (to A.S.N.). S.H.H. gratefully acknowledges infrastructure funding from Canadian Foundation for Innovation (CFI) CFI LOF Project 30100, and the University of Calgary's Canada First Research Excellence Fund (CFREF) program, the Global Research Initiative (GRI) in Sustainable Low Carbon Unconventional Resources.

## Author contributions

H.Z. designed, developed the experiments, and characterized the experimental data in this project. H.Z., S.H.H., and A.S.N. conceptualized, analyzed and interpreted data, and wrote the paper. All authors provided feedback on the final version of the paper.

## Competing interests

The authors declare no competing interests.
