## [Peer Review File · Nature Communications]

Self-assembly of highly ordered micro- and nanoparticle depositsREVIEWER COMMENTS

Reviewer #1 (Remarks to the Author):

The manuscript proposed a novel method to enable the suppression of coffee-ring effect by combining a hydrophilic surface and neutral wetting mold. This leads to a uniform particle coating, which is important for a wide range of applications including ink-jet printing and bio-sensors. In the manuscript, the authors show that 1. the particle-laden drop bounded by the mold leaves the uniform deposition (fig. 2); 2. the particle deposition can be monolayer under certain physical parameter range (e.g. particle concentration) (fig. 3). In a second step, the authors investigate the coating of two different size beads (200 and 1000 nm) with a desired (tree-leaf) pattern and discuss the capability as microfilters (fig.4). Finally, the authors discuss the theoretical interpretation behind the evaporation process of the liquid film bounded by the mold (fig.5).

The proposed method is interesting and will be a nice addition in the field of "suppression of coffee-ring effect". However, I have the following essential concerns for both of the experimental and theoretical parts;

With regard to experimental part: the interaction between the liquid film and mold is a main determinant of the final pattern of coating, but this is not enough-characterized. I would like to see the roughness of the inner wall of the mold and the value of contact angle (if possible for contact angle). For complementing this, it would be useful to measure the fluid flow during the evaporation process using PIV (e.g. see S.F.Shimobayashi et al., Sci. Rep. 8, 17769, (2018)). The authors will see almost zero radial flow if the contact angle is close to $\pi/2$.

With regard to theoretical part: the authors define t_{Diff} in Eq. 1 as R^2/D . The authors use the diffusion constant (D) as the diffusion constant by thermally-driven Brownian motion; however, I think that this is not right. This should be the diffusion constant by capillary flow created by spatially heterogenous evaporation rate. So, beyond this point, I think that the theoretical model needs to be re-considered.

I also have the following specific points:

1. Fig.1(e) has too much information for the first figure so that it would not be easy to understand.
2. Fig.2(c-e) should have error bars.
3. Fig.3 is the result from only one film. It would be nice to add the statical data.
4. Why are the enlarged images of top/bottom layers in Fig.3(c) not shown? They look non-uniform for me. Given that the authors use a confocal microscope for imaging, it is easy to show and gives a direct sense for pores in the film as well. Related to this, it is not clear how the pore size in Fig.3(e) was estimated. I would think that it would be better to directly estimate from the confocal images rather than using the equation $(\alpha_{sl} \times PS_s + (1-\alpha_{sl}) \times PS_{sl})$. I also do not see the values of PS_s and PS_{sl} in the text.

Reviewer #3 (Remarks to the Author):

The manuscript describes a novel method to avoid coffee-ring effects in evaporating particle-laden drops and obtain a uniform particle deposition out of it. The nature-inspired method makes use of a confining mold with neutral wettability which prevents the formation of significant curvature at the liquid/air/mold interface and thus the establishment of capillary flow and consequent non-uniform particle distribution. The method is admittedly easy, intuitive, and the evidence presented by the authors in favor of its practical applicability is appropriate, including accurate measurements of particle distribution, multi-layer deposits with size-dependent particle segregation, and pore density gradients (the authors also use the technique to fabricate electrodes, although this appears only in the Methods section). The authors also propose (and fully derive in SI) a theoretical model that supposedly subsumes most of the phenomenology they evidence in a few parameters, such as contact angle and Peclet number.

The manuscript is overall clearly if redundantly written, well structured, and it provides an excellent bibliography on coffee ring literature and particularly methods to avoid non-uniform particle deposition (also in Table S1). The supporting material mainly focuses on coffee-ring-like depositions in absence of mold, and includes 4 videos as well.

The idea presented by the authors is simple, intuitive, and worth adding to the vast literature on the subject to the benefit of both academic and applied scientists.

However, in current form there are a few points that deserve further attention from the authors before a complete assessment of the merits of the paper can be made:

1) The main point is that the experimental validation of the theoretical model is largely missing. The authors identify a parameter window whereby the preferred result (uniform particle deposition) is expected to happen, and provide experiments to confirm only that (page 18; the three points mentioned in the text are by the way not reported in the phase diagram of figure 5d). Therefore, while that single prediction seems preliminarily confirmed, the rest of the phase diagram lacks experimental confirmation. The authors should provide experimental data to prove or disprove it. This would give a solid base for their qualitative claims for e.g. what would happen when θ is larger or smaller than 90 deg.

In fact, we don't fully agree with the explanations given by the authors for those cases (which are admittedly outside the interest and preference of the authors, yet necessary to make the model convincing), and only an experimental validation would clear out the situation. For instance, for θ smaller than 90 deg (which is the case represented in figure 5a), one could instead expect coffee ring formation along the sidewall of the mold, due to the local divergence of the evaporation flux at the receding contact line in that situation. Is this seen in experiments? Would that eventually translate in non-uniform particle distribution on the substrate after full solvent evaporation? Moreover, the curved drop interface for $\theta < 90$ deg, with center thinner than the edge, could induce break-up of the droplet upon contact of the solvent/air interface with the substrate (i.e., film rupture) and hence again a non-uniform particle pattern deposited.

2) The explanation of particle self-sorting (segregation) for the multi-particle deposition case (figure 4) seems also not entirely convincing. The authors seem to claim that only the larger (1 μ m diameter) particles tend to stick to the free solvent interface, and hence end up in the upper layer upon solvent evaporation. One could argue that the same if not a higher tendency is shown by the smaller (200 nm) particles, since they have higher mobility (as the authors remind) and experience higher free energy reduction in sticking to the solvent/air interface than the larger particles.

3) Figure 4b and insets of figure 4c seem to show streaming patterns in what look like non-uniform particle distribution patterns. If that is the case, how does this match with the authors' claim for uniform particle deposition? Do the authors see this also in multi-particle experiments conducted in the circular mold?

Our impression is that, even under $\theta = 90$ deg, the non-circular geometry (e.g., non-constant curvature) of the mold itself may induce local evaporation flux gradients along the contact line, and hence local capillary flows that would alter particle distribution. The authors are encouraged to disprove this hypothesis, or accordingly revise the generality of the mold shape-independence of their technique.

4) We note that the presence of pore size gradients (page 14) is a noteworthy yet not unique feature of the present technique, rather it is quite ubiquitous in coffee-ring literature, given the inherent size-sorting of polydisperse suspensions during solvent evaporation and/or contact line receding.

5) The main text contains perhaps too many re-statements of the advantages of the technique, and generally could benefit from a more concise exposition. It also contains several typos and grammatical hiccups, and some non-defined acronyms (e.g, RSD on page 4).

Also, Figure 1b does not really seem to show uniform salt deposition given the evident presence of cracks.

Response to referees' comments

Response to Reviewer #1

Q1) The proposed method is interesting and will be a nice addition in the field of “suppression of coffee-ring effect”. However, I have the following essential concerns for both of the experimental and theoretical parts; With regard to experimental part: the interaction between the liquid film and mold is a main determinant of the final pattern of coating, but this is not enough-characterized. I would like to see the roughness of the inner wall of the mold and the value of contact angle (if possible for contact angle). For complementing this, it would be useful to measure the fluid flow during the evaporation process using PIV (e.g. see S.F.Shimobayashi et al., Sci. Rep. 8, 17769, (2018)). The authors will see almost zero radial flow if the contact angle is close to $\pi/2$.

We appreciate the valuable comments on the effect of wall roughness and fluid flow measurement using PIV. To address the concerns, we measured the contact angle between the fluid and the mold wall (Figure S2 supplementary). The contact angle in the proposed method is measured to be 90° , minimizing the interaction between the liquid film and the wall. We agree that surface roughness is also a key parameter in the liquid-wall interaction. We used atomic force microscopy (AFM) to analyze the roughness and topography of PSA wall (Figure R1). The average roughness of PSA wall is measured to be around 56 nm. Studies have shown that surface roughness may come to play at contact angles deviated from 90° , wherein increased roughness of a surface intensifies its hydrophilic or hydrophobic property [1,2]. Hence, we did not see the effect of the measured roughness on the liquid film-PSA wall interactions.

Figure R1. The topography and roughness of the PSA wall measured using atomic force microscopy.

We also used microparticle image velocimetry (μ PIV) to visualize particle motions when the film evaporates under the no-coffee-ring condition. The results confirm the elimination of radial motility of particles tested under the proposed method (Figure R2). Tracking of particles over time depicts that the net radial movement of particles is almost zero (Figure R2a), with their small fluctuation during evaporation (Figure R2b).

We also examined whether the wall shape (circular or rectangular patterns) affects particle movement during evaporation. No difference between these two patterns is recorded (Movies S5 and S6, Supplementary Information), which indicates that the wall curvature does not affect the evaporation rate.

Figure R2. Particle tracking during evaporation using the μ PIV device. a) Snapshots of the particles in the evaporating film over time, representing the negligible radial motility of particles; scale bars are 100 μm . b) The descending trajectory of particles during film evaporation, where the particles follow almost a vertical line.

Figures R1, R2, and μ PIV movies, along with their discussion, are added to the manuscript [refer to SI-Figure S3 and Figure S6].

[1] Hay, K. M., M. I. Dragila, and J. Liburdy. "Theoretical model for the wetting of a rough surface." *Journal of colloid and interface science* 325.2 (2008): 472-477.

[2] Whyman, Gene, and Edward Bormashenko. "How to make the Cassie wetting state stable?." *Langmuir* 27.13 (2011): 8171-8176.

Q2) With regard to theoretical part: the authors define t_{Diff} in Eq. 1 as R^2/D . The authors use the diffusion constant (D) as the diffusion constant by thermally-driven Brownian motion; however, I think that this is not right. This should be the diffusion constant by capillary flow created by spatially heterogenous evaporation rate. So, beyond this point, I think that the theoretical model needs to be re-considered.

Thanks for the comment. We acknowledge that the migration of particles cannot be solely modeled by diffusion. In fact, during contact line pinning, the particles move to the peripheral part due to the capillary-driven advective flux and not diffusion. The advection time (t_{Adv}) can be scaled by R/\dot{E} , where R is the hydrodynamic radius of the mold and \dot{E} is the evaporation rate. The advection build-up of the particles at the contact line will be warded off by diffusional backflow owing to the development of particle's concentration gradient. The diffusion time (t_{Diff}) can also be scaled by R^2/D . Assuming the use of spherical particles at a uniform temperature, possessing the same kinetic energy in a quiescent fluid, the stokes-Einstein equation is considered for diffusion coefficient (D). Therefore, we revised the scaling and normalized the pinning time with $\Delta t = t_{Adv} - t_{Diff}$. Knowing that the diffusion scale is much less than the advection scale, and considering the Peclet number as $Pe = \dot{E}H/D$, the pinning time can be approximated as:

$$\left| \frac{t_{pin}}{\Delta t} \right| = \begin{cases} \frac{H}{R} \frac{\delta\theta}{Pe(1 + \sin\theta)^2} & \theta > 90^\circ \\ \frac{H}{R} \frac{\delta\theta \sin\theta}{Pe(1 + \sin\theta)^2} & \theta < 90^\circ \end{cases} \quad (1)$$

The discussion about this change is added in the supplementary information (section: Theoretical model of the proposed method-equation S4) and the main text of the manuscript (Page 16-equation 1-Figure 5).

Q3) Fig.1(e) has too much information for the first figure so that it would not be easy to understand.

Thanks for the comment. We modified the figure and only put the essential explanations, including (1) oxygen plasma of substrate that leads to superhydrophilic substrate; (2) Confining the substrate with a proper mold ($\theta \approx 90^\circ$); (3) Optimization of liquid volume; (4) The balanced rate of evaporation that leads to uniform distribution of particles.

Q4) Fig.2(c-e) should have error bars.

Agreed. The error bars are now added to Figures 2 (c-e).

Q5) Fig.3 is the result from only one film. It would be nice to add the statical data.

We appreciate the comment. The roughness value with the confidence interval of 95% is added to the caption of Figure 3.

Q6) Why are the enlarged images of top/bottom layers in Fig.3(c) not shown? They look non-uniform for me. Given that the authors use a confocal microscope for imaging, it is easy to show and gives a direct sense for pores in the film as well. Related to this, it is not clear how the pore size in Fig.3(e) was estimated. I would think that it would be better to directly estimate from the confocal images rather than using the equation $(\alpha_{sl} \times PS_s + (1-\alpha_{sl}) \times PS_{sl})$. I also do not see the values of PS_s and PS_{sl} in the text.

Thanks for the comment. We agree that the particles' final patterns are not highly uniform. This could be because we used a dilute solution to ensure the stratification of polystyrene particles with a small transition zone. A dilute particle-laden solution reduces particle-particle interactions due to an increase in particles' mean free path, consequently resulting in size-driven stratification of particles. A more uniform particle distribution is achievable using higher particle concentrations (please see Figure R3).

In the dilute solution, the slightly non-uniform distribution is inevitable due to Bénard Marangoni instabilities. This instability emerges from local gradients in the interfacial tension along the water-vapor interface of the evaporating film, resulting in the formation of patterns [1-5]. This instability occurs in diluted particle-laden liquid where the particles at the interface do not form a rich mono/multi-layer. In the increased concentration of particles, the film drying culminates in a uniform pattern (Figures 2b and 2d in the main text), where the particle concentration is high enough to cover the whole liquid/air interface. When it comes to stratification, increasing the concentration of particles may prohibit particles from moving easily along the film height due to particle-particle interaction, leading to an increase in the

transition zone (Figure R3a). In such a case, particle stratification occurs, although not perfectly (Figure R3b).

Figure R3. Stratification of the 1000 nm and 200 nm polystyrene particles with the concentration of $\phi_0=0.1$ and with the initial film thickness of approximately $H=500$ nm. a) 3-dimensional confocal microscopy image of the particles, representing the extended transition zone with respect to the case with $\phi_0=0.01$. b) The bottom layer (left) and the top layer (right) of the stratified deposition, representing the uniform deposition of the particles. Scale bars:500 μm .

We agree with the reviewer that a direct estimation of pore sizes from images is much better; however, the pore sizes are in the nanometer scale (≈ 130 nm-190 nm). Therefore, we did not have enough resolution from confocal images to accurately capture the pore size. Hence, we used $\alpha_{sl} \times PS_s + (1 - \alpha_{sl}) \times PS_l$ formula to calculate the pore size. Assuming a uniform arrangement of particles, the PS_s and PS_l values in $\alpha_{sl} \times PS_s + (1 - \alpha_{sl}) \times PS_l$ are calculated theoretically to be approximately 46 nm and 231 nm, respectively. We now added the pore sizes in the revised manuscript.

The descriptions about the limitation of the method are now added to the main text (Pages 13-14). Figure R3 is also added to the supplementary information (Figure S9).

- [1] Nguyen, Van X., and Kathleen J. Stebe. "Patterning of small particles by a surfactant-enhanced Marangoni-Bénard instability." *Physical Review Letters* 88.16 (2002): 164501.
- [2] Bassou, N., and Yahya Rharbi. "Role of Benard– Marangoni instabilities during solvent evaporation in polymer surface corrugations." *Langmuir* 25.1 (2009): 624-632.
- [3] H. Machrafi, A. Rednikov, P. Colinet, P. Dauby, Bénard instabilities in a binary-liquid layer evaporating into an inert gas, *Journal of colloid and interface science*, 349 (2010) 331-353.
- [4] H. Machrafi, A. Rednikov, P. Colinet, P.C. Dauby, Bénard instabilities in a binary-liquid layer evaporating into an inert gas: Stability of quasistationary and time-dependent reference profiles, *The European Physical Journal Special Topics*, 192 (2011) 71-81.
- [5] Bormashenko, Edward, et al. "Mesoscopic patterning in evaporated polymer solutions: New experimental data and physical mechanisms." *Langmuir* 21.21 (2005): 9604-9609.

Response to Reviewer #3

Q1) The main point is that the experimental validation of the theoretical model is largely missing. The authors identify a parameter window whereby the preferred result (uniform particle deposition) is expected to happen, and provide experiments to confirm only that (page 18; the three points mentioned in the text are by the way not reported in the phase diagram of figure 5d). Therefore, while that single prediction seems preliminarily confirmed, the rest of the phase diagram lacks experimental confirmation. The authors should provide experimental data to prove or disprove it. This would give a solid base for their qualitative claims for e.g. what would happen when θ is larger or smaller than 90 deg. In fact, we don't fully agree with the explanations given by the authors for those cases (which are admittedly outside the interest and preference of the authors, yet necessary to make the model convincing), and only an experimental validation would clear out the situation. For instance, for θ smaller than 90 deg (which is the case represented in figure 5a), one could instead expect coffee ring formation along the sidewall of the mold, due to the local divergence of the evaporation flux at the receding contact line in that situation. Is this seen in experiments? Would that eventually translate in non-uniform particle distribution on the substrate after full solvent evaporation? Moreover, the curved drop interface for $\theta < 90$ deg, with center thinner than the edge, could induce break-up of the droplet upon contact of the solvent/air interface with the substrate (i.e., film rupture) and hence again a non-uniform particle pattern deposited.

We really appreciate the reviewer's comment, and we agree that experimental results on contact angles other than 90° can shed light on the validity of the theoretical model. We did further experiments to characterize the coffee-ring behavior where the contact angle between the film and the mold's wall deviates from 90° (hydrophilic and hydrophobic walls). As per reviewer's comment, we monitored the deposition patterns of particles during evaporation and evaluated the impinging force to examine to what extent the theoretical model matches the experimental data. PDMS molds with tunable wettability (using oxygen plasma) and similar roughness among the tests were selected to perform this set of experiments [1]. Plasma treatment renders an initially hydrophobic PDMS to hydrophilic. We ensured that the time of the experiment was short enough (<20 min) so that the contact angle variation was negligible. Figure R4 shows patterns of particles for various contact angles and impinging fluxes.

Figure R4. Two-dimensional (2D) phase diagram showing the experimental and theoretical monolayer, coffee-ring, and coffee-eye areas as a function of \bar{F}_{impinge} and θ . The \bar{F}_{impinge} and θ values for the experimental data are: 1) $\bar{F}_{\text{impinge}} \approx 0.72$, $\theta \approx 54^\circ$; 2) $\bar{F}_{\text{impinge}} \approx 0.9$, $\theta \approx 90^\circ$; 3) $\bar{F}_{\text{impinge}} \approx 0.86$, $\theta = 106^\circ$; 4) $\bar{F}_{\text{impinge}} \approx 0.6$, $\theta \approx 90^\circ$; 5) $\bar{F}_{\text{impinge}} \approx 0.42$, $\theta \approx 54^\circ$; 6) $\bar{F}_{\text{impinge}} \approx 0.35$, $\theta \approx 90^\circ$; 7) $\bar{F}_{\text{impinge}} \approx 0.5$, $\theta \approx 106^\circ$.

The patterns of particles obtained from new experimental data show that coffee-eye pattern occurs when the wall contact angle is smaller than 90° . For the PDMS wall with the contact angle $<90^\circ$ (here $\theta = 54^\circ$), in some cases (e.g., case 5), liquid rupturing leads to the creation of a possibly random hollow feature (uncoated) in the middle of the substrate, as the reviewer suggested. We also observed some particles

attaching to the PDMS wall; however, the intensity of deposited particles on the wall is negligible compared to those deposited on the substrate. Therefore, the occurrence of coffee-eye is more highlighted as the substrate is superhydrophilic ($\theta < 5^\circ$). For the systems with $H \gg R$, however, the concentration of deposited particles on the wall may not be negligible. For the PDMS molds with the hydrophobic property (here $\theta = 106^\circ$), a high percentage of particles deposit on the edge of the substrate (cases 3 and 7), as confirmed by the theoretical model.

The additional experimental points (cases 1 to 7 in Figure R4) demonstrate that the theoretical model could predict the coffee ring/eye behavior presented in this work. However, we admit that other complexities may contribute to changing the final pattern of particles and need further experimentations.

Figure 5d in the main text is updated with an explanation to address the concern (Pages 20-21).

[1] Tan, Say Hwa, et al. "Oxygen plasma treatment for reducing hydrophobicity of a sealed polydimethylsiloxane microchannel." *Biomicrofluidics* 4.3 (2010): 032204.

Q2) The explanation of particle self-sorting (segregation) for the multi-particle deposition case (figure 4) seems also not entirely convincing. The authors seem to claim that only the larger (1 μm diameter) particles tend to stick to the free solvent interface, and hence end up in the upper layer upon solvent evaporation. One could argue that the same if not a higher tendency is shown by the smaller (200 nm) particles, since they have higher mobility (as the authors remind) and experience higher free energy reduction in sticking to the solvent/air interface than the larger particles.

We acknowledge that there is also a good chance for smaller particles (200 nm) traveling toward the interface due to higher mobility and reduction of Gibbs free energy. Indeed, literature has shown that particle stratification is not only a function of particle size but also evaporation rate, temperature, and fluid height and viscosity, generally the Peclet number of the particle. In some cases, larger particles end up in the upper layer upon solvent evaporation [1, 2]. This work focuses on the concept of particle stratification in small systems, while selectivity on particle size organization can be performed as described in the literature [1, 2]. For instance, if the Peclet values of both particles are larger than 1, the small particles segregate in a layer on top of the larger ones; and if the Peclet values of the large and small particles are higher than and smaller than 1, respectively, the small particles form the bottom layer [1,2].

The descriptions about different arrangements in particles stratifications are now added to the main text (Page 14).

[1] Trueman, R. E., et al. "Auto-stratification in drying colloidal dispersions: A diffusive model." *Journal of colloid and interface science* 377.1 (2012): 207-212.

[2] Fortini, Andrea, et al. "Dynamic stratification in drying films of colloidal mixtures." *Physical review letters* 116.11 (2016): 118301.

Q3) Figure 4b and insets of figure 4c seem to show streaming patterns in what look like non-uniform particle distribution patterns. If that is the case, how does this match with the authors' claim for uniform particle deposition? Do the authors see this also in multi-particle experiments conducted in the circular mold?

Thanks for the comment. We agree that the particles' final patterns are not highly uniform. This could be because we used a dilute solution to ensure the stratification of polystyrene particles with a small transition zone. A dilute particle-laden solution reduces particle-particle interactions due to an increase in particles' mean free path, consequently resulting in size-driven stratification of particles. A more uniform particle distribution is achievable using higher particle concentrations (please see Figure R5).

In the dilute solution, the slightly non-uniform distribution is inevitable due to Bénard Marangoni instabilities. This instability emerges from local gradients in the interfacial tension along the water-vapor interface of the evaporating film, resulting in the formation of patterns [1-5]. This instability occurs in diluted particle-laden liquid where the particles at the interface do not form a rich mono/multi-layer. In the increased concentration of particles, the film drying culminates in a uniform pattern (Figures 2b and 2d in the main text), where the particle concentration is high enough to cover the whole liquid/air interface. When it comes to stratification, increasing the concentration of particles may prohibit particles from moving easily along the film height due to particle-particle interaction, leading to an increase in the transition zone (Figure R5a). In such a case, particle stratification occurs, although not perfectly (Figure R5b).

Figure R5. Stratification of the 1000 nm and 200 nm polystyrene particles with the concentration of $\phi_0=0.1$ and with the initial film thickness of approximately $H=500$ nm. a) 3-dimensional confocal microscopy image of the particles, representing the extended transition zone with respect to the case with $\phi_0=0.01$. b) The bottom layer (left) and the top layer (right) of the stratified deposition, representing the uniform deposition of the particles. Scale bars: 500 μm .

The descriptions about the limitation of the method are now added to the main text (Pages 13-14). Figure R5 is also added to the supplementary information (Figure S9).

[1] Nguyen, Van X., and Kathleen J. Stebe. "Patterning of small particles by a surfactant-enhanced Marangoni-Bénard instability." *Physical Review Letters* 88.16 (2002): 164501.

[2] Bassou, N., and Yahya Rharbi. "Role of Benard- Marangoni instabilities during solvent evaporation in polymer surface corrugations." *Langmuir* 25.1 (2009): 624-632.

[3] H. Machrafi, A. Rednikov, P. Colinet, P. Dauby, Bénard instabilities in a binary-liquid layer evaporating into an inert gas, *Journal of colloid and interface science*, 349 (2010) 331-353.

[4] H. Machrafi, A. Rednikov, P. Colinet, P.C. Dauby, Bénard instabilities in a binary-liquid layer evaporating into an inert gas: Stability of quasistationary and time-dependent reference profiles, *The European Physical Journal Special Topics*, 192 (2011) 71-81.

[5] Bormashenko, Edward, et al. "Mesoscopic patterning in evaporated polymer solutions: New experimental data and physical mechanisms." *Langmuir* 21.21 (2005): 9604-9609.

Q4) Our impression is that, even under $\theta = 90$ deg, the non-circular geometry (e.g., non-constant curvature) of the mold itself may induce local evaporation flux gradients along the contact line, and hence local capillary flows that would alter particle distribution. The authors are encouraged to disprove this hypothesis, or accordingly revise the generality of the mold shape-independence of their technique.

We appreciate this comment and the suggestion. We agree that a non-circular curvature may lead to a local evaporation gradient. In the case of the flat meniscus (no curvature in liquid/air interface), where we have uniform evaporation at the interface, the effect of contact line-driven evaporation gradient may become less significant. To confirm this hypothesis, we traced the movement of particles during evaporation using a μ PIV instrument and compared the results for the round and rectangular patterns (Movies S5 and S6 in supplementary information). The results show that the particles for both patterns only move vertically down over time with no net lateral movement during evaporation, confirming that the evaporation rate is almost constant along the interface.

We also monitored the deposition of CNHs nanoparticles in square and A-like patterns to compare their deposition pattern in both curved and straight walls (Figure R6). The results show that as long as we use a mold with $\theta = 90^\circ$ and incorporate a hydrophilic substrate, the particles deposit in a uniform pattern, showing the flexibility of this method.

Figure R6. Deposition of CNHs nanoparticles on different patterns. Uniform deposition of particles on a) square shape and b) A-like shape, representing the negligible effect of wall curvature on the evaporation rate. a) Scale bars: subsection II is 500 μm and subsections III and IV are 100 μm . b) Scale bars: 100 μm .

The description above is added to the main text (Page 10) along with Figure R6 (Figure S7), added in SI.

Q5) We note that the presence of pore size gradients (page 14) is a noteworthy yet not unique feature of the present technique, rather it is quite ubiquitous in coffee-ring literature, given the inherent size-sorting of polydisperse suspensions during solvent evaporation and/or contact line receding.

As mentioned by the reviewer, various studies in the literature addressed particle size sorting, but almost all the experimental/theoretical studies have been performed by focusing on a macro-scale. Using the valuable information on macro-scale particle sorting from the literature [1-4] and a dearth of study on small scale particle stratification, we performed particle sorting in a small-scale pattern in this work applying the proposed molding technique, with the vision of opening a new avenue for downstream studies on small-scale variable pore size filters.

[1] Trueman, R. E., et al. "Auto-stratification in drying colloidal dispersions: A diffusive model." *Journal of colloid and interface science* 377.1 (2012): 207-212.

[2] Trueman, R. E., et al. "Autostratification in drying colloidal dispersions: experimental investigations." *Langmuir* 28.7 (2012): 3420-3428.

[3] Fortini, Andrea, et al. "Dynamic stratification in drying films of colloidal mixtures." *Physical review letters* 116.11 (2016): 118301.

[4] Nikiforow, Irina, et al. "Self-stratification during film formation from latex blends driven by differences in collective diffusivity." *Langmuir* 26.16 (2010): 13162-13167.

Q6) The main text contains perhaps too many re-statements of the advantages of the technique, and generally could benefit from a more concise exposition. It also contains several typos and grammatical hiccups, and some non-defined acronyms (e.g, RSD on page 4).

Many thanks for the comment. Considering the new experimental data, we revised several sentences throughout the manuscript to delineate the utility scope of this work. We tried to do a concise discussion on the physical phenomena and removed unnecessary statements. We also did a proofread to correct the typos and grammatical errors.

Q7) Also, Figure 1b does not really seem to show uniform salt deposition given the evident presence of cracks.

Thanks for the comment. The cracks on the lake substrate are mainly owing to the dried mud and barely due to the non-uniform deposition. Dry lakes are usually covered with shallow water during the rainy season, where the layer of water is thin and move around the dry lake-bed on windy days. Consequently, an extremely hard and smooth substrate may develop. This results in a "cracked-mud" surface and teepee structure desiccation features.

REVIEWER COMMENTS

Reviewer #1 (Remarks to the Author):

The paper is improved after the revision.
I have two questions/comments.

(1) μ PIV looks cool. Can the authors show the analysis till the evaporation ends? I think that the particle movement at the final moment is critical for supporting the author's conclusion, i.e., meniscus-free uniform deposition.

(2) I am still confused about the pore size calculation. The author's equation is only true for the closest packing. Looking at Fig.4c, it seems to me that the calculation is not scientifically true. Also, Fig. R3 is hard to see.

Reviewer #3 (Remarks to the Author):

The authors have proposed a thorough revision of their original submission which incorporates the received feedback. The new submission is consequently significantly improved and expanded both in the main text and in the supplementary information, whereby additional experiments and characterizations have fortified the validity of the proposed method and its theoretical rationalization.

We consider the current manuscript suitable for publication in the journal, minus a few residual typos and possibly a rephrasing of line 11-13 of the Abstract.

Response to referees' comments

Reviewer #1 (Remarks to the Author):

The paper is improved after the revision.
I have two questions/comments.

(1) μ PIV looks cool. Can the authors show the analysis till the evaporation ends? I think that the particle movement at the final moment is critical for supporting the author's conclusion, i.e., meniscus-free uniform deposition.

We appreciate the reviewer's comment. We further used μ PIV to trace the particles during the evaporation for a longer time (see Figure R1). Given that the particles move downward vertically and knowing that μ PIV cannot present the 3D movement of particles (i.e. it only captures the overlap of particles in a 2D plane), tracing the particles using μ PIV until the last steps of evaporation is challenging. During the evaporation, some particles go out of focus while new particles enter the μ PIV focusing zone (see the cloudy regions in Figure R1). However, we showed the dynamics of particle deposition in supplementary information using bright field microscopy, indicating the particle movement during the evaporation and their final deposited pattern (Supplementary movie S2 and snapshots in Figure 2b).

Figure R1: μ PIV image sequences of the suspended particles during the evaporation

(2) I am still confused about the pore size calculation. The author's equation is only true for the closest packing. Looking at Fig.4c, it seems to me that the calculation is not scientifically true. Also, Fig. R3 is hard to see.

We agree that the method applied in this study for calculating pore size is only valid for close-packing particles. Therefore, we rationally employed a known approach used for fracture-matrix systems in the literature [1]. In this approach, the pore size distribution is calculated, considering gaps (fractures) between the close-packing particles (matrix) in each scanned layer. We characterized the surface area and radius of the fractures (referred to as PS_f) and used this information to determine the areal fraction of close-packing particles (γ). The value γ is then used to calculate a weighted average of close-packing particle pore size ($\alpha_{sl} PS_s + (1 - \alpha_{sl}) PS_l$) and fracture radius (PS_f). This pore size is introduced as the average pore size of each layer (see equation R1).

$$PS_{av} = \gamma[\alpha_{sl} PS_s + (1 - \alpha_{sl}) PS_l] + (1 - \gamma)PS_f \quad \text{Equation (R1)}$$

We believe this approach presents a more accurate pore size calculation for the systems with fractures. It is worth mentioning that γ value depends on the concentration of the particles, wherein by the increase in the concentration of particles, γ value tends to unity and the average pore size approaches the close-packing pore size. Figure R2 shows the average pore size of less dense and dense (close-packing) systems at each layer. The shaded area highlights the pore size difference between the close-packing system and the less dense one. Figure 4e in the revised manuscript is replaced with Figure R2, and the above information is added to the text to address the concern.

Figure R2. The average nanoscale pore size of the deposited fine- and micro-particles in close-packing and less dense packing states, showing the pore size gradient throughout the layers.

Regarding Figure R3, increasing the concentration of particles complicates distinguishing particles using confocal microscopy, and only a general trend of particles' location is observable. However, we modified the figure to present a sharper view of particles.

[1] Tiab, Djebbar, and Erle C. Donaldson. Petrophysics: theory and practice of measuring reservoir rock and fluid transport properties. Gulf professional publishing, 2015.

Reviewer #3 (Remarks to the Author):

The authors have proposed a thorough revision of their original submission which incorporates the received feedback. The new submission is consequently significantly improved and expanded both in the main text and in the supplementary information, whereby additional experiments and characterizations have fortified the validity of the proposed method and its theoretical rationalization.

We consider the current manuscript suitable for publication in the journal, minus a few residual typos and possibly a rephrasing of line 11-13 of the Abstract.

Many thanks for the comment. We revised lines 11-13 in the abstract and fixed the typos.

REVIEWERS' COMMENTS

Reviewer #1 (Remarks to the Author):

I am ok with the experimental parts now.
For the theoretical part, I leave the decision to the additional assigned reviewer working on the theory.